# Recent Advances in Application of Computer-Aided Drug Design in Anti-Influenza A Virus Drug Discovery

**DOI:** 10.3390/ijms23094738

**Published:** 2022-04-25

**Authors:** Dahai Yu, Linlin Wang, Ye Wang

**Affiliations:** Key Laboratory for Molecular Enzymology and Engineering of Ministry of Education, School of Life Sciences, Jilin University, 2699 Qianjin Street, Changchun 130012, China; yudahai@jlu.edu.cn (D.Y.); linlinw20@mails.jlu.edu.cn (L.W.)

**Keywords:** anti-influenza A virus, drug discovery, computer-aided drug design (CADD), small-molecule inhibitor

## Abstract

Influenza A is an acute respiratory infectious disease caused by the influenza A virus, which seriously threatens global human health and causes substantial economic losses every year. With the emergence of new viral strains, anti-influenza drugs remain the most effective treatment for influenza A. Research on traditional, innovative small-molecule drugs faces many challenges, while computer-aided drug design (CADD) offers opportunities for the rapid and effective development of innovative drugs. This literature review describes the general process of CADD, the viral proteins that play an essential role in the life cycle of the influenza A virus and can be used as therapeutic targets for anti-influenza drugs, and examples of drug screening of viral target proteins by applying the CADD approach. Finally, the main limitations of current CADD strategies in anti-influenza drug discovery and the field’s future directions are discussed.

## 1. Introduction

Influenza is an infectious disease caused by influenza viruses, killing 300,000 to 500,000 people worldwide every year [1,2]. The notorious “Spanish flu” of 1918, caused by the H1N1 virus, caused 50 million deaths worldwide [3], and the “swine flu”, caused by the H1N1 virus in 2009, caused at least 16,000 deaths [4]. The influenza virus is one of the primary human pathogens that cause respiratory diseases. It causes high morbidity and mortality through seasonal influenza and global pandemics.

Influenza viruses are negative-strand RNA viruses (−ssRNA viruses) belonging to the family Orthomyxoviridae [5]. The viruses can be classified into type A, type B, and type C according to the antigenic differences between their nucleoprotein and matrix protein. Influenza A and B viruses account for most influenza globally, and the influenza A virus is highly contagious and can cause pandemics. The genome of influenza A consists of eight RNA segments. Segments 1, 2, and 3 encode RNA-dependent RNA polymerase (EC 2.7.7.48) (RdRp, composed of PB2, PB1, and PA), segment 4 is responsible for encoding hemagglutinin (HA); segment 5 is responsible for encoding nuclear protein (NP), segment 6 encodes neuraminidase (EC 3.2.1.18) (NA), segment 7 encodes matrix protein (M2 and M1), and segment 8 encodes nuclear export protein (NEP) and nonstructural protein 1 (NS1) genes [6,7]. Influenza A viruses are categorized by the surface glycoproteins HA and NA subtypes, with 16 HA (H1–H16) and 9 NA (N1–N9) antigenic subtypes having been identified [8,9].

For the prevention and treatment of influenza, vaccines and antiviral drugs are still the most effective options. In terms of reducing morbidity and mortality, the vaccine is considered the most desirable method of controlling influenza virus pandemics [10,11]. However, the influenza vaccine is only effective if the vaccine strain matches the prevalent influenza strain. It is a challenge each year to determine the composition of the influenza vaccine months in advance to allow time for the manufacture and distribution of the vaccine.

Drugs designed based on the specific targets of influenza viruses are considered to be an effective option for the treatment of influenza. These target proteins are involved in essential functions at different stages of the influenza virus life cycle. Table 1 shows the FDA-approved antiviral drugs available for the treatment of influenza virus infections, including PA endonuclease inhibitors (Baloxavir acid) [12], neuraminidase inhibitors (Oseltamivir and Zanamivir) [13,14], and M2 ion channel inhibitors (Adamantanes and Rimantadine) [15,16].

However, flaws in the drugs and the rapid emergence of drug-resistant mutations in pandemic and seasonal influenza viruses urgently require the development of new antivirals. Computer-aided drug design (CADD) has been increasingly used to develop potential drugs to prevent or treat influenza. CADD comprises computer technology, protein structures, mathematical modeling, and statistics to understand and predict the binding patterns and energies of small-molecule inhibitors of potential targets [17,18,19,20,21,22]. Applying CADD techniques in drug development can reduce drug development costs and shorten the drug development cycle compared to traditional drug development. This review discusses the structure and function of specific targets involved in the influenza virus life cycle. We provide an overview of recent advances in CADD application examples, including the strategies of computational structure-based drug design (SBDD) and ligand-based drug design (LBDD) for developing druggable target inhibitors (Figure 1). Moreover, close attention is paid to the essential role of computational techniques in overcoming drug resistance.

## 2. Influenza A Virus Life Cycle and Protein

The life cycle of an influenza virus is a complex biological process that can be divided into the following steps (Figure 2). (i) Virion attaches to the host cell surface via the influenza virus surface glycoprotein HA; (ii) influenza virus M2 protein creates an ion channel that regulates the pH value of the cell membrane and triggers the fusion between the viral envelope and the endosomal membrane, releasing the viral ribonucleoprotein (vRNP) complex into the cytoplasm; (iii) the vRNP complex is transported to the nucleus, and RNA polymerase (PB2, PB1, and PA) begins transcription and replication of viral RNA; (iv) the newly synthesized viral surface proteins are transported to the cell surface via the Golgi apparatus, while the newly synthesized nuclear proteins are transported back to the nucleus for reassembly into the vRNP complex; and (v) viral surface proteins are assembled into viral progeny. The viral progeny is finally budded and released from the cell membrane by NA, which cleaves sialic acid on the surface of the host cell membrane. Almost any viral protein of the life cycle steps can become a potential therapeutic target to control and prevent influenza infection [23,24,25]. Then, these viral proteins can be used for drug screening in a subsequent CADD approach.

## 3. The Application of Computer-Aided Drug Design (CADD)

### 3.1. Structure-Based Drug Design (SBDD)

SBDD is a technique based on receptor structure that has proven useful for rapidly identifying biologically active hit compounds in early drug discovery. The receptor is usually a protein, and its three-dimensional crystal structure can be obtained by searching protein databases, such as the RCSB PDB database (https://www.rcsb.org/ accessed on 1 April 2022) [26]. However, the structure can also be constructed by the homology modeling method (https://swissmodel.expasy.org/ accessed on 1 April 2022) in the case of an unknown protein structure [27]. The artificial intelligence system developed by DeepMind in 2021, AlphaFold, is a system that can predict the 3D structure of a protein from its amino acid sequence [28]. It can be accurate to the extent that it competes with experiments. Subsequently, David Baker’s team at the University Of Washington School Of Medicine successfully developed a tool based on deep learning that can quickly and accurately predict the structure of target proteins based on limited information, RoseTTAFold [29]. Force field–based molecular dynamics (MD) simulations provide a dynamic view of the protein structure over time, accompanied by optimization of the protein conformation to bring the protein system to an equilibrium state [30,31]. It also reveals the time-dependent dynamics of change of ligand–protein binding sites in the system [32]. Commonly used MD simulation programs are GROMACS [33], AMBER [34], NAMD [35], CHARMM [36], OpenMM [37], etc. GROMACS is the most commonly used and efficient molecular dynamics package, supporting many common classical force fields such as AMBER, CHARMM, GROMOS, and OPLS with simple operation and rich functions for simulation of protein–ligand complexes. Amber is more professional in ligand parameterization, but the force field is singular (only the Amber force field), and the calculation speed is not complimentary. NAMD supports CHARMM, Amber, and GROMACS force fields and shows strong advantages when coupled with VMD software, but its analysis tools are relatively lacking. CHARMM is a widely recognized and applied molecular dynamics simulation program; the charmm force field is well suited to simulate the thermodynamic properties of organic small molecules, but the software is too complex to operate. OpenMM is an open-source high-performance toolkit for molecular simulations. Selecting the appropriate library of the small-molecule database is another key to structure-based drug design that greatly improves efficiency [38]. Commonly used databases are ZINC [39], a free commercial compound database for virtual screening of over 750 million purchasable compounds; PubChem [40], the world’s largest collection of freely accessible chemical information to search for chemical substances by name, molecular formula, structure, and other identifiers; DrugBank [41], a comprehensive, free-to-access online database containing information on drugs and drug targets; ChEMBL [42] is a manually curated database of bioactive molecules with drug-like properties. It brings together chemical, bioactivity, and genomic data to aid the translation of genomic information into effective new drugs. These compound libraries vary in size and are either free or commercially available. Researchers can select different databases or filter collections depending on their purpose.

### 3.2. Ligand-Based Drug Design (LBDD)

LBDD is also known as indirect drug design [43]. The structure–activity data of a set of known active molecules are used to construct a structure–activity relationship or pharmacophore model when the three-dimensional structure of the protein target is not available, including pharmacophore model [44], quantitative structure–activity relationship (QSAR) [45], molecular shape-based superposition [46], and other specific methods. The appropriate research method depends on the amount of available information. The more accurate information is, the more reliable the results will be.

### 3.3. Virtual Screening

In molecular docking–based virtual screening techniques, protein–ligand complexes are assigned a score that correlates with the predicted binding affinity, which can be calculated from a physics-based, empirical, or knowledge-based potential function [47,48]. Commonly used virtual screening programs are AutoDock [49], AutoDock Vina [50], MOE [51], GLIDE [52], Discovery Studio [53], etc. The online server PAINS-Remover is designed and constructed to remove pan assay interference compounds (PAINS) from screening libraries and exclude them from biology assays [54]. The SwissADME website predicts the physicochemical description of compounds online, including absorption, distribution, metabolism, elimination, and toxicity [55]. These virtual screening and compound optimization filtering tools are necessary to improve drug development’s success rate and reduce the problem of wasted funds in the later stages of drug development.

### 3.4. Biology Assay

A follow-up biology assay is crucial in CADD applications to help calibrate theoretical results with experimental results. Experimental evaluation/validation includes experiments at the molecular level, cellular level, animal level, pharmacokinetics, etc. Each experiment level is analyzed, and the compounds that perform well in each experiment are chosen to be drug candidates.

## 4. Function and Structure of Target Proteins for Anti-Influenza Virus Candidates

### 4.1. PA Endonuclease

The RNA-dependent RNA polymerase (RdRp) of influenza A virus, which is essential for viral RNA transcription and replication, is a large (250 kDa) heterotrimeric complex composed of three subunits: PB1, PB2, and PA (Figure 3) [56]. Once in the host cell’s nucleus, the RdRp begins transcription and subsequent replication of viral RNA (vRNA) [57]. During the transcription of vRNA, the cap-binding domain in PB2 captures the host pre-mRNA essential for the virus RNA transcription. The N-terminal of PA contains an endonuclease center that can cleavage the pre-mRNA 8–14 nt after the M7GTP cap [58]. After the cleavage, the cap of the pre-mRNA is used as primers for viral mRNA synthesis with polymerase catalytic site in PB1 [59,60]. Thus, four targets in RdRp were used to develop the new anti-influenza drugs, including the cap-dependent endonuclease domain in PA, cap-binding domain in PB2, the interaction domain between PA and PA, and PB1, the interaction domain between the PB1 and PB2. A PA endonuclease inhibitor that binds into the active site pocket can block the host RNA cleavage, which can inhibit the transcription of the viral genome [61]. Since the catalytic residues in the cleavage reaction are conserved, it suggests that PA endonuclease inhibitors may be effective against different influenza virus subtypes.

Although the structure of PA endonuclease has been reported previously, no structural data indicate the molecular mechanism at high resolution. Recent studies by Fan et al. gave the complete three-dimensional crystal structure of the RdRp from strain A/duck/Fujian/01/2002(H5N1) (PDB ID 6QPF) [62]. Moreover, Omoto et al. provided insight into the crystal structure and molecular details of the interaction of the PA endonuclease structural domain from strain Influenza A/California/04/2009 (pH1N1) with baloxavir acid (PDB ID 6FS6) [63]. In this structure, baloxavir acid binds at the active site of PA endonuclease with residues Tyr24, Lys34, Ala37, Ile38, His41, Glu80, Asp108, Glu119, Tyr130, and Lys134. These crystal structures provide a basis for the design of structure-based specific influenza antiviral drug development.

Recent screenings by Zhang et al. of plant extracts were used to screen the inhibitor against the PA endonuclease and its mutant. After the primary screening in vitro and virtual screening in silico, bilobetin was identified as able to competitively inhibit the PA endonuclease [64]. Similarly, Meng et al. virtually screened the drug lifitegrast, which effectively inhibits PA and mutant PA-I38T endonuclease activity by a drug repurposing approach. Gel-based PA nuclease analysis determined that the IC_50_ of lifitegrast for PA and PA-I38T endonuclease were 32.82 ± 1.34 µM and 26.81 ± 1.2 µM, respectively [65]. Zhang et al. identified a raltegravir derivative as a potential novel PA endonuclease inhibitor by 3D-QSAR modeling and a docking-based virtual screening approach [66]. Pala et al. proposed a coupled pharmacophore/docking virtual screening method to identify inhibitors with PA endonuclease inhibitory activity in a PA enzymatic assay and observed antiviral activity in the low micromolar range in a cell-based influenza virus assay [67]. Moreover, Ferro et al. developed a three-dimensional pharmacophore model and obtained three “hit compounds” through virtual screening. The binding poses of these hit compounds were investigated by molecular docking method and enzymatic analysis using recombinant PA endonuclease. Subsequently, the compound that most effectively inhibited the endonuclease reaction was selected with an IC_50_ of 12 μM [68].

**Figure 3 ijms-23-04738-f003:**
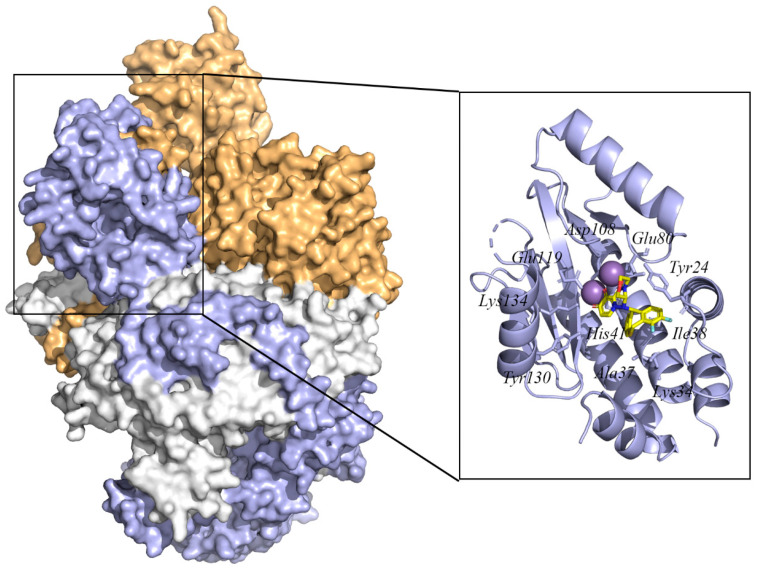
The surface structure of the influenza A virus RNA-dependent RNA polymerase complex (PB2, PB1, and PA) (PDB ID: 6QPF) [62] and the cartoon structure of PA endonuclease with inhibitor baloxavir acid (PDB ID: 6FS6) [63]. In the polymerase complex, the PA domain is colored blue, the PB1 domain is colored white, and the PB2 domain is colored orange. The residues in the PA endonuclease active site are indicated by blue sticks. The baloxavir acid is colored yellow. The Mn ions are indicated with violet spheres. The protein structure was displayed and analyzed through the PyMOL visualization program [69,70,71].

### 4.2. Hemagglutinin

The hemagglutinin (HA) of influenza A virus, which is essential for viral entry and uncoating, is a homotrimer commonly divided into a head and stem regions (Figure 4). Each chain is synthesized as a precursor polypeptide (HA0) and then cleaved into two fragments, HA1 (328 amino acids) and HA2 (221 amino acids), linked by a disulfide bond [72,73]. After attaching to the host receptor, the influenza viral particles are transported to the endosome via endocytosis. The pH in the endosome becomes acidic by proton entry [74]. The acidic pH triggers viral–host membrane fusion that is mediated by conformational rearrangements in the HA [75]. The prerequisite for such conformational rearrangements is the proteolytic processing of the HA. HA is translated as a single polypeptide chain HA0, which is then cleaved by host proteases into the HA1 and HA2 subunits. The membrane fusion machinery is encoded mainly by HA2, while HA1 is entirely responsible for receptor binding. The overall structure of uncleaved HA0 is almost identical to the cleaved HA [76]. This metastable conformation is poised for low pH–induced structural rearrangements to accomplish viral–host membrane fusion.

Since the complex structure of HA was solved, the stem region on HA represented a promising target for future influenza antiviral development [77]. Interestingly, it was found to be the target site of the Russian-made broad-spectrum antiviral drug arbidol [78]. Kadam et al. determined crystal structures of arbidol in complex with influenza virus HA from Influenza A virus strain (A/Shanghai/02/2013(H7N9)) (PDB ID 5T6S) [79]. The inhibitor binding site is located at the interface between adjacent protomers of the HA2 trimer and is composed of residues Arg54, Leu55, Glu57, Lys58, Thr59, and Asn60 from HA2 and residues Glu90’, Tyr94’, Glu97’, Leu98’, and Ala101’ from adjacent HA2.

**Figure 4 ijms-23-04738-f004:**
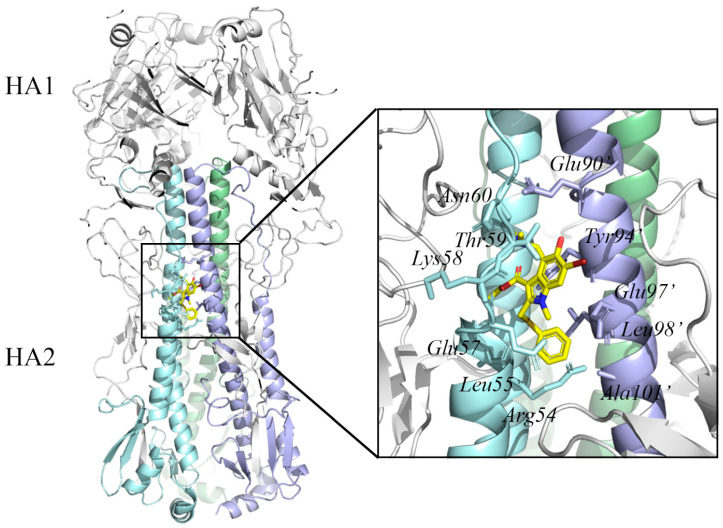
Structure of influenza A virus hemagglutinin in complex with the antiviral drug arbidol (PDB ID: 5T6S) [79]. The HA1 trimer molecules (head) are represented as a gray cartoon, and HA2 trimer molecules (stem) are represented as cyan, green, and blue cartoons, respectively. Arbidol is shown as yellow sticks. The residues in the arbidol binding site are indicated by sticks. The protein structure was displayed and analyzed through the PyMOL visualization program [69,70,71].

Bodian et al. used computer-aided methods to screen putative HA inhibitors and identified benzoquinone and hydroquinone as effective fusion inhibitors of HA [80]. Chang et al. analyzed over 200,000 compounds from the United States National Cancer Institute (NCI) database and identified potential HA inhibitors by computational strategy. It revealed that compound NSC85561 showed significant antiviral activity with EC50 values ranging from 2.31 to 2.53 µM and negligible cytotoxicity (CC50 > 700 µM) [81]. Waldmann et al. describe a trivalent glycopeptide mimetic, a nanomolar multivalent ligand, which binds to avian influenza hemagglutinin H5 by in silico design, chemical synthesis, and binding analysis [82].

### 4.3. Neuraminidase

Influenza A virus neuraminidase (NA), also known as sialidase, is a tetramer made up of four identical subunits, each of which has a sialic-acid binding pocket (Figure 5A) [83]. Many studies believe that the influenza A virus NA facilitates the release of viral progeny and the migration of the virus into the respiratory tract during the life cycle of the virus [84,85]. The virus buds from the host cell as the viral genome and capsid proteins assemble into new influenza virus particles. However, between the mature influenza virus and the host cell, the sialic acid residue at the end of the HA molecule is stably attached to the glycosyl group on the surface of the HA receptor molecule via a 2–6 or 2–3 glycosidic bond, preventing the immediate separation of the influenza virus from the host cell [86]. NA has exo-glycan hydrolase activity and can cleave the α-ketosidically linked Neu5Ac residues covering the ends of various glycoconjugates [87]. Therefore, the mature virus particles eventually detach from the host cells and infect epithelial cells, leading to the further spread of the influenza virus in the patient. In addition, the function of NA that cleaves the terminal α-ketoside-linked Neu5Ac residue appears to be conserved in the influenza A virus [88]. It suggests that NA inhibitors may be effective in the treatment of the influenza A virus.

By analyzing the crystal structure of NA, we can obtain the tetrameric structure of NA and the binding site of sialic acid. Then, through the conformational modification of sialic acid and a series of experimental verifications, we have obtained effective influenza drugs such as oseltamivir, which is the first influenza drug developed by the CADD method [13]. As previous studies found that influenza A NA is divided into two distinct groups, group-1 and group-2. Group-1 contains N1, N4, N5, and N8 subtypes, while group-2 contains N2, N3, N6, N7, and N9. In this review, the crystal structure of N1 from H5N1 avian influenza A virus strain as group-1 (PDB ID: 2HU4) [89], while the crystal structure of N9 from Influenza A virus strain (A/Shanghai/02/2013(H7N9)) as group-2 (PDB ID: 5L15) [90]. The crystal structure shows that group 1 NA has a distinct 150-cavity formed by a 150-loop (Figure 5B,C). Despite the slight difference in conformation, the residues in the active site centers of the two groups of NA are relatively conserved. The active site of N1 contains residues Glu119, Gln136, Val149, Asp151, Arg156, Glu276, and Tyr347. Meanwhile, the active site of N9 contains residues Glu119, Gln136, Ile149, Asp151, Arg156, Glu276, and Asn347.

Although the anti-influenza drugs oseltamivir and zanamivir, which target the viral NA, are currently approved by the World Health Organization (WHO), many NA inhibitors are still being studied for better efficacy [91]. Several studies have shown insufficient oral bioavailability of zanamivir, while resistance to oseltamivir has been developed in Europe due to clinical overuse since 2007 [92,93,94]. As fungal secondary metabolites that have potent antiviral activity against various known pathogenic viruses have been reported in many studies, Md. Mukthar Mia used a systematic screening method to find the top candidates of natural fungal compounds for neuraminidase inhibitors and predict potential drug candidates against the influenza A virus [95]. In his work, molecular docking methods, computational pharmacokinetics, and pharmacology were developed, which can help search for inhibitor candidates. The compounds sitaxentan, ergoloid mesylate, capecitabine, and fenoterol were found to be the candidates against neuraminidase. Zhao et al. discovered a novel lead compound **6a** by ligand-based virtual screening, receptor-based virtual screening, molecular dynamics simulation (MD), and bioassay validation (IC_50_ = 7.10 ± 0.2 μM). Then, a series of novel acyl hydrazone NA inhibitors 6b–6g were designed and synthesized based on the lead compound **6a**. Compound 6e exerts the most potency, with an IC_50_ value of 2.37 ± 0.5 μM against NA, which is lower than that of oseltamivir carboxylate (IC_50_ = 3.84 μM) [96]. Additionally, through pharmacophore-based virtual screening and molecular dynamic simulation, Zhong et al. discovered a new lead compound **4** (ZINC01121127). Some novel thiophene ring-containing NA inhibitors were synthesized by optimizing the backbone of the lead compound **4**. Compound **4b** showed the strongest inhibitory activity against NA (IC_50_ = 0.03 μM), which was superior to the positive control oseltamivir (IC_50_ = 0.06 μM) [97]. To analyze the correlation between NA and diarylheptanes, Yoo et al. used chromatographic techniques to isolate four new compounds from the rhizomes of A. officinarum and 26 known diarylheptanes to build a small-molecule database. Then, a structure-based virtual screen (SBVS) was performed to identify compounds that could stably bind NA and validate these in in vitro NA enzyme assays. Molecular docking was used to investigate the binding mechanism of inhibitors of NA [98].

### 4.4. Matrix Protein M2

The matrix protein M2 of type A influenza virus, which plays a crucial role in viral uncoating, is a homotetramer composed of four transmembrane peptides arranged across the membrane’s viral lipid in an N-out C-in manner [99,100,101] (Figure 6). When the influenza A virions enter the host cell via endosome, the matrix protein M2, a highly selective proton channel activated at low pH, helps H+ ions enter the virion’s interior [102,103,104,105,106]. The inward proton via the matrix protein M2 leads to the acidification of the viral interior. Therefore, the protein–protein interaction is damaged especially in the case between M1 protein and viral ribonucleoproteins. Then the uncoating of the virions in the endosome is facilitated. As a result, RNA proteins are released from the endosomes into the cytoplasm and transported to the nucleus.

The crystal structures of the rimantadine-bound M2 TM domain (residues 22–46) in both conformational states, the inward-closed state (PDB ID 6BKL) and the inward-open state (PDB ID 6BOC), were resolved [107]. At the N-terminal end of the M2 TM structural domain, the channel pore is restricted by a hydrophobic gate formed by the Val27 side chain. There are five pore-facing residues in the M2 TM structural domain, which are Val27, Ala30, Ser31, Gly34, His37, and Trp41. In particular, His37 is oriented towards the center of the channel, where it acts as a selectivity filter and proton shuttle, giving M2 a strong proton selectivity in both inward-closed and inward-open states.

Although blockers that target M2 ion channels, including amantadine and rimantadine, are World Health Organization (WHO) approved drugs for the treatment of viral infections [91], ineffective treatment was due to adamantine scarcity, lower efficacy, and adverse effects on the central nervous system with the rapid development of resistance [108,109]. Radosevic et al. used ligand-based virtual screening and molecular docking to search for candidate anti-influenza ion channel inhibitors for wild-type and adamantine-resistant influenza A viruses. As a result, guanethidine was the best-ranked drug selected from ligand-based virtual screening, and the experiment proved its measurable anti-influenza activity [110]. Duncan et al. identified chebulagic acid as an inhibitor of the M2 for the treatment of influenza A from two natural product libraries by virtual screening approach. It was also shown that chebulagic acid selectively restored the growth of yeast expressing the M2 mutant (S31N). Molecular docking elucidates the interaction mechanism of the chebulagic acid hydrolysis fragment with the M2 mutant (S31N) amino acid residue, while in vitro experiments demonstrated that chebulagic acid inhibits influenza A virus replication [111].

## 5. Future Perspectives and Conclusions

In spite of the challenge of producing and distributing vaccines matched to pandemic influenza strains several months in advance each year, antiviral drugs appear to be effective in controlling seasonal pandemics, especially in the early stages of rapid transmission. Viral proteins that play critical steps in the viral life cycle could be potential therapeutic targets for influenza treatment, including PA endonuclease, hemagglutinin, neuraminidases, and matrix protein M2. CADD, the most critical technology in current drug development, has undoubtedly accelerated the development of influenza drugs. It provides computational tools and algorithms that save time and costs and reduce the risk of detecting infeasible development routes. We show various examples of CADD strategies used to develop small-molecule inhibitors of the influenza virus. These examples are mainly related to structure-based drug design and ligand-based drug design in small-molecule inhibitor development, including virtual screening, 3D-QSAR, molecular dynamics, pharmacokinetic calculations, etc. One of the most successful cases is the neuraminidase inhibitor oseltamivir, which was developed based on the neuraminidase substrate sialic acid structure. Although several potent influenza drugs have been approved for marketing by the FDA, the emergence of drug-resistant strains has made the drugs resistant to the influenza virus. Therefore, there is an urgent need to find new mechanisms and methods to combat the ever-mutating influenza viruses.

One approach is that we need to find novel drug targets. The PA endonuclease inhibitor, Baloxavir acid, launched in 2018, broke the dominance of neuraminidase as an influenza drug target. This reminds us that some overlooked influenza viral proteins may also be the star targets for influenza treatment, such as NP, NEP, NS1, etc. Moreover, the combination of drugs with different target mechanisms may overcome the challenge of drug resistance.

In addition, with the improvement of computing power and algorithm development, multi-scale computer methods will further optimize the efficiency of virtual drug screening. As a powerful data mining tool, artificial intelligence (AI) techniques have been implemented in the field of drug design [29,112]. AlphaFold and RoseTTAFold’s accurate protein structure prediction solves the challenge of protein full structure resolution. This exciting shift in structural informatics will accelerate our more profound research into human health and medicine based on protein structure. Another critical method, in silico virtual screening, has been constructed for many successful applications, but the method still has many apparent limitations. For example, most docking methods consider only binding affinity and ignore other parameters, so the docking score is not a desirable indicator of drug efficacy. In addition, the fact that the false-positive rate of molecular docking based virtual screening is high should not be ignored [113]. A novel strategy to develop reliable protein–ligand scoring functions by augmenting the traditional scoring function Vina score with correction terms (OnionNet-SFCT), which significantly enhances the AutoDock Vina prediction capability [114]. More software tools are combined strategies for training and filtering small-molecule databases using deep learning methods [115,116,117].

In conclusion, CADD technology provides an essential tool for identifying new lead compounds, thus accelerating the development of novel antiviral drugs for influenza. In the future, CADD technology relying on artificial intelligence is expected to be more intelligent in covering all aspects of new drug discovery and development, and we look forward to witnessing a revolution in new drug discovery.

## Figures and Tables

**Figure 1 ijms-23-04738-f001:**
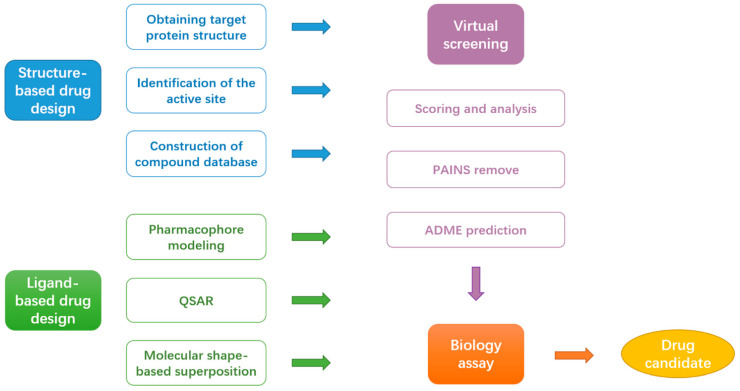
Flow chart of commonly used computer-aided drug design (CADD) approaches.

**Figure 2 ijms-23-04738-f002:**
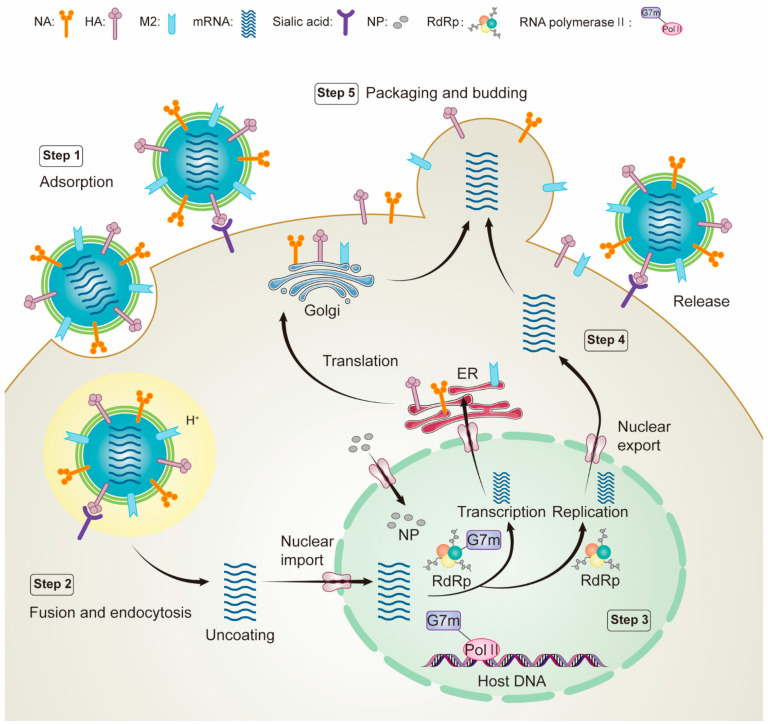
Schematic diagram of influenza virus life cycle: step 1, virus attaches to the host cell surface via the influenza virus surface glycoprotein hemagglutinin; step 2, influenza virus M2 protein produces ion channels that regulate the pH of the cell membrane, trigger fusion and release of viral ribonucleoprotein (vRNP) complexes into the cytoplasm; step 3, the vRNP complex is transported to the nucleus and RNA polymerase (PB2, PB1, and PA) begins transcription and replication of viral RNA; step 4, The newly synthesized viral surface proteins are transported to the cell surface via the Golgi apparatus, while the newly synthesized nuclear proteins are transported back to the nucleus for reassembly into the vRNP complex; and step 5, viral surface proteins are assembled into viral progeny. The viral progeny are finally budded and released from the cell membrane by NA, which cleaves sialic acid on the host cell membrane surface.

**Figure 5 ijms-23-04738-f005:**
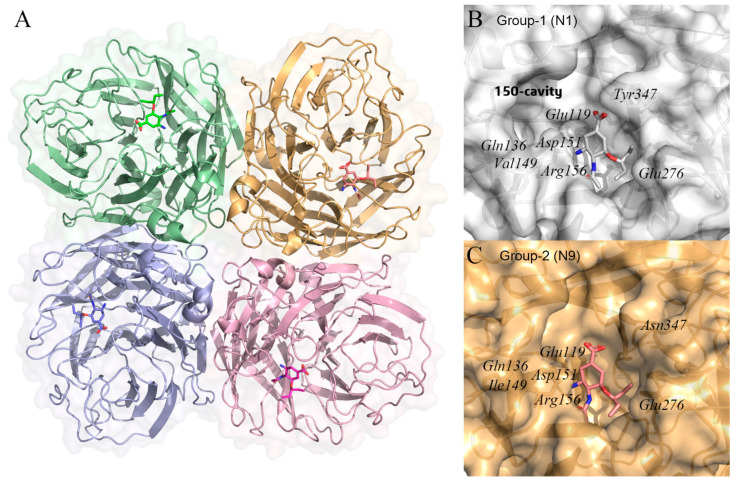
(**A**) structures of the influenza A virus neuraminidase tetramer with oseltamivir (PDB ID: 2HU4) [89]. Each monomer in the neuraminidase tetramer is represented as a different colored surface. Oseltamivir binds to the active site of neuraminidase and is shown in stick form. (**B**,**C**) molecular surfaces representation of group-1 (N1) (PDB ID: 2HU4) and group-2 (N9) neuraminidases (PDB ID: 5L15) [90] with oseltamivir. In group-1 neuraminidase has a distinct 150-cavity formed by 150-loop. The protein structure was displayed and analyzed through the PyMOL visualization program [69,70,71].

**Figure 6 ijms-23-04738-f006:**
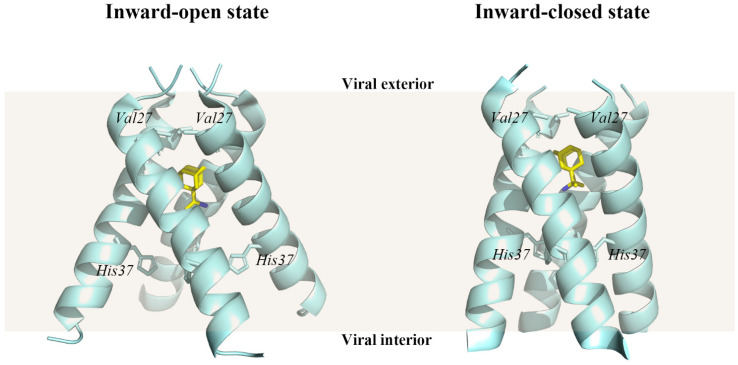
Structures of the influenza A virus M2 protein channel in complex with rimantadine. The transmembrane (TM) structural domain of M2 bound to rimantadine in the inward-open state (PDB ID: 6BOC) and in the inward-closed state (PDB ID: 6BKL) [107]. Wheat shade represents the viral membrane. The protein structure was displayed and analyzed through the PyMOL visualization program [69,70,71].

**Table 1 ijms-23-04738-t001:** Influenza A virus genome segments encode protein, function, and targeted drugs.

Segments	Protein	Function	Targeted Drugs	Refs.
1	polymerase PB2	Viral RNA transcription and replication	NA	
2	polymerase PB1	Viral RNA transcription and replication	NA	
3	polymerase PA	Viral RNA transcription and replication	Baloxavir acid	[12]
4	hemagglutinin	Interacts with sialylated glycan (sialoside) receptors on the surface of the host cell; the virus fuses with the cell membrane, allowing the virus to enter the cell.	NA	
5	nuclear protein	Combines with three polymerase subunits to form ribonucleoprotein (RNP)	NA	
6	neuraminidase	Cleavage of sialic acid receptors to release progeny viruses from host cells	Oseltamivir; Zanamivir	[13,14]
7	matrix protein M2	Ion (Na^+^) channels and regulates the pH value of the cell membrane; virus assembly and budding	Amantadine; Rimantadine	[15,16]
	matrix protein M1	The structural component of virion; forming a virus endoskeleton to mediate virus assembly	NA	
8	nuclear export protein (nonstructural protein 2)	Mediating the nuclear export of viral ribonucleoprotein (RNP) complexes	NA	
	nonstructural protein 1	Blocks the nuclear export of host-encoded immune factor messenger RNA	NA	

## Data Availability

Not applicable.

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
