# Peer review of "Recent Advances in Application of Computer-Aided Drug Design in Anti-Influenza A Virus Drug Discovery"

_ijms, 2022, doi:10.3390/ijms23094738_

Round 1

Reviewer 1 Report

Here, Yu et al., the authors reviewed the recent progress in the development of anti-influenza A virus drugs using computer-aided drug design (CADD). This submission is within the scope of the International Journal of Molecular Sciences and the topic is interesting.  In my view, the manuscript has the novelty necessary to justify a new publication. I recommend minor revision.

I based my decision on the following issues.

1) Throughout the text, the authors mentioned several enzymes. Please, add the enzyme classification number for all enzymes in the manuscript. The authors should include the EC number the first time the enzyme is mentioned in the text. For instance, in section 1.1, the authors say: 
"Segments 1, 2, and 3 encode RNA-dependent RNA polymerase
(RdRp, composed of PB2, PB1, and PA)..."

It should be as follows.

"Segments 1, 2, and 3 encode RNA-dependent RNA polymerase (EC 2.7.7.48) (RdRp, composed of PB2, PB1, and PA)..."

The authors should follow this procedure for all enzymes mentioned in the manuscript. The authors can easily find enzyme classification numbers (EC numbers) and related biochemical information in the following databases: https://www.uniprot.org/, https://www.brenda-enzymes.org/index.php, and https://enzyme.expasy.org/.

2) Throughout the text, the authors mentioned several PDB IDs, e.g., the legend in figure 3. It is necessary to add the main reference paper for each structure.

3) In the legends of figures 3-6, it is necessary to add the name of the program used to generate them and the reference of this program.

4) In the introduction section, the authors said:
CADD comprises computer technology, protein crystallization, mathematical model and statistics to understand and predict the binding patterns and energies of small molecule inhibitors to potential targets [17-19].

It is not only crystallographic data the source of 3D models for CADD. And definitely, I wouldn't refer to it as protein crystallization only. You may rewrite this sentence as follows.

CADD comprises computer technology, protein structures, mathematical model, and statistics to understand and predict the binding patterns and energies of small molecule inhibitors to potential targets [17-22].

Also, add the following references.
20.  https://pubmed.ncbi.nlm.nih.gov/17348837/
21. https://pubmed.ncbi.nlm.nih.gov/17348836/
22. https://pubmed.ncbi.nlm.nih.gov/13679062/

5) Please renumber the references to accommodate the new ones.

6) In the section named "5. Future Perspectives and Conclusion", the authors said:
As a powerful data mining tool, artificial intelligence (AI) techniques have been implemented in the field of drug design.

Please add the following references related to this topic.
https://pubmed.ncbi.nlm.nih.gov/34282049/
https://pubmed.ncbi.nlm.nih.gov/32410551/

7) In summary, this work reaches the high-quality standards necessary for a publication in the International Journal of Molecular Sciences. I recommend accepting it after minor revision.

Author Response

Dear Editor and Reviewers,

We would like to thank the reviewers for their critical examination of our manuscript. Their comments have been constructive for revising the manuscript. The issues they raised, and our responses, are outlined below. The corrected portions in the text are highlighted in the revised manuscript.

I hope it will be suitable for publication now.

Thank you,

Ye Wang

22 Apr 2022

Response to the comments from the reviewer 1:

Here, Yu et al., the authors reviewed the recent progress in the development of anti-influenza A virus drugs using computer-aided drug design (CADD). This submission is within the scope of the International Journal of Molecular Sciences and the topic is interesting. In my view, the manuscript has the novelty necessary to justify a new publication. I recommend minor revision.

Response: Thank you.

I based my decision on the following issues.

> 1) Throughout the text, the authors mentioned several enzymes. Please, add the enzyme classification number for all enzymes in the manuscript. The authors should include the EC number the first time the enzyme is mentioned in the text. For instance, in section 1.1, the authors say: "Segments 1, 2, and 3 encode RNA-dependent RNA polymerase (RdRp, composed of PB2, PB1, and PA)..." It should be as follows. "Segments 1, 2, and 3 encode RNA-dependent RNA polymerase (EC 2.7.7.48) (RdRp, composed of PB2, PB1, and PA)..." The authors should follow this procedure for all enzymes mentioned in the manuscript. The authors can easily find enzyme classification numbers (EC numbers) and related biochemical information in the following databases: https://www.uniprot.org/, https://www.brenda-enzymes.org/index.php, and https://enzyme.expasy.org/.

Response: We are grateful for the suggestion. As suggested by the reviewer, we have added the EC number of the enzymes RNA-dependent RNA polymerase and neuraminidase (Page 1, Lines 34 and 36).

> 2) Throughout the text, the authors mentioned several PDB IDs, e.g., the legend in figure 3. It is necessary to add the main reference paper for each structure.

Response: We are grateful for the suggestion. As suggested by the reviewer, we have added the reference paper for each structure in the figures.

>3) In the legends of figures 3-6, it is necessary to add the name of the program used to generate them and the reference of this program.

Response: We are grateful for the suggestion. As suggested by the reviewer, we have added the program's name (PYMOL) and the reference 69-71.

>4) In the introduction section, the authors said:
CADD comprises computer technology, protein crystallization, mathematical model and statistics to understand and predict the binding patterns and energies of small molecule inhibitors to potential targets [17-19].

It is not only crystallographic data the source of 3D models for CADD. And definitely, I wouldn't refer to it as protein crystallization only. You may rewrite this sentence as follows.

CADD comprises computer technology, protein structures, mathematical model, and statistics to understand and predict the binding patterns and energies of small molecule inhibitors to potential targets [17-22].

Also, add the following references.
20.  https://pubmed.ncbi.nlm.nih.gov/17348837/
21. https://pubmed.ncbi.nlm.nih.gov/17348836/
22.
https://pubmed.ncbi.nlm.nih.gov/13679062/

Response: We are grateful for the suggestion. We rewrote the sentence, and references 20-22 has been added in the revised manuscript.

> 5) Please renumber the references to accommodate the new ones.

Response:  Thank you for reminding us. We have renumbered the references.

> 6) In the section named "5. Future Perspectives and Conclusion", the authors said: As a powerful data mining tool, artificial intelligence (AI) techniques have been implemented in the field of drug design.

Please add the following references related to this topic.
https://pubmed.ncbi.nlm.nih.gov/34282049/
https://pubmed.ncbi.nlm.nih.gov/32410551/

Response: Thank you for the suggestion. The recommend references have been added in the revised manuscript (Page 12, Line 393, references 29 and 112).

>7) In summary, this work reaches the high-quality standards necessary for a publication in the International Journal of Molecular Sciences. I recommend accepting it after minor revision.

Response: Thank you.

Reviewer 2 Report

Presented is a review of drug design efforts targeting the Influenza A

type virus. The presentation is well focused on improtant viral

proteins and information on known drugs is appropriately

presented. While a broad overview of CADD methods is not needed,

expansion of the description of CADD methods, tools for MD simulations

and databases would be helpful to the reader.  Specific comments are

below.

Figure 2: Typo in the figure "Trascraption"

line 86: Typo "silicic acid"

lines 116-117: The following statement needs to be clarified as the

ligand binding site is typically known before performing MD

simulations. "It also reveals the binding site of the ligand when the

ligand is bound to the protein in the system [29]."

line 119 etc. Expanding the the MD simulation packages (CHARMM, NAMD

and OpenMM) as well as the force fields that are available would be

useful to readers with some description of their scope. This should

include small molecule force fields appropriate for drug design.

line 120 and onward. A better description of the databases of

commercially available compounds is needed as this is important

informtion for the reader.

line 125: It is stated that LBDD was used before AI, but AI is simply

LBDD (or SBDD) that utilizes neutral nets to perform the model

development rather than more simple regression models.

line 136: Typo "AutDock" is used twice.

line 138: "pan-detection interfering compounds" to "pan assay interference compounds"

When the authors cite other studies they often use the full name of

the first author, for example "Maggie C Duncan et al." Just the last

name is required "Duncan et al."

Author Response

Dear Editor and Reviewers,

We would like to thank the reviewers for their critical examination of our manuscript. Their comments have been constructive for revising the manuscript. The issues they raised, and our responses, are outlined below. The corrected portions in the text are highlighted in the revised manuscript.

I hope it will be suitable for publication now.

Thank you,

Ye Wang

22 Apr 2022

Response to the comments from the reviewer 2:

Presented is a review of drug design efforts targeting the Influenza A type virus. The presentation is well focused on important viral proteins and information on known drugs is appropriately presented. While a broad overview of CADD methods is not needed, expansion of the description of CADD methods, tools for MD simulations and databases would be helpful to the reader. Specific comments are below.

Response: Thank you.

>Figure 2: Typo in the figure "Trascraption"

Response: We are extremely grateful to the reviewer for pointing out this problem. We have revised it in figure 2.

>line 86: Typo "silicic acid"

Response: We are extremely grateful to the reviewer for pointing out this problem. We have modified this word (Page 3, Line 84 and Page 4, Line 98).

>lines 116-117: The following statement needs to be clarified as the ligand binding site is typically known before performing MD simulations. "It also reveals the binding site of the ligand when the ligand is bound to the protein in the system [29]."

Response: We sincerely appreciate the reviewer's suggestion. According to the reviewer's comment, we change the sentence to "It also reveals the time-dependent dynamics of change of ligand-protein binding sites in the system" on Page 5, Line 115.

>line 119 etc. Expanding the the MD simulation packages (CHARMM, NAMD and OpenMM) as well as the force fields that are available would be useful to readers with some description of their scope. This should include small molecule force fields appropriate for drug design.

Response: Thank you for your precious comments. We have extended the description of the MD simulation package (Page 5, Line 116 to 128):

"Commonly used MD simulation programs are GROMACS [33], AMBER [34], NAMD [35], CHARMM [36], and OpenMM [37] etc. GROMACS is the most commonly used and efficient molecular dynamics package, supporting many common classical force fields such as AMBER, CHARMM, GROMOS, and OPLS, with simple operation and rich functions for simulation protein-ligand complexes. Amber is more professional in ligand parameterization, but the force field is single (only Amber force field), and the calculation speed is not complimentary. NAMD supports CHARMM, Amber, and GROMACS force fields and shows strong advantages when coupled with VMD software, but its analysis tools are relatively lacking. CHARMM is a widely recognized and applied molecular dynamics simulation program; charmm force field is well suited to simulate the thermodynamic properties of organic small molecules, but the software is too complex to operate. OpenMM is an open-source high-performance toolkit for molecular simulations."

And the references have been adjusted accordingly (references 35, 36 and 37).

>line 120 and onward. A better description of the databases of commercially available compounds is needed as this is important information for the reader.

Response: Thank you for your precious comments. We have extended the description of databases of commercially available compounds as follows, and the references adjusted accordingly (Page 5, Line 129 to 137):

"Commonly used databases are ZINC [39], a free commercial compound database for virtual screening of over 750 million purchasable compounds; PubChem [40], the world's largest collection of freely accessible chemical information to search for chemical substances by name, molecular formula, structure, and other identifiers; DrugBank [41], a comprehensive, free-to-access, online database containing information on drugs and drug targets; ChEMBL [42], a manually curated database of bioactive molecules with drug-like properties. It brings together chemical, bioactivity, and genomic data to aid the translation of genomic information into effective new drugs, etc."

>line 125: It is stated that LBDD was used before AI, but AI is simply LBDD (or SBDD) that utilizes neutral nets to perform the model development rather than more simple regression models.

Response: We thank the reviewer for this critical question. We have modified this expression according to the comment on Page 5, Line 141.

>line 136: Typo "AutDock" is used twice.

Response: Thank you for your comments. Sorry for the mistakes, we have revised “AutDock” to “AutoDock” (Page 5, Line 152).

>line 138: "pan-detection interfering compounds" to "pan assay interference compounds"

Response: We are grateful to the reviewer for pointing out this issue. We have modified the expression to "pan assay interference compounds" on Page 5, Line 154.

>When the authors cite other studies they often use the full name of the first author, for example "Maggie C Duncan et al." Just the last name is required "Duncan et al."

Response: Thank you for your precious comments. When citing studies of other authors, we normalized the expressions.
